# Measuring Chronic Stress in Broiler Chickens: Effects of Environmental Complexity and Stocking Density on Immunoglobulin-A Levels

**DOI:** 10.3390/ani13132058

**Published:** 2023-06-22

**Authors:** Andrew M. Campbell, Mallory G. Anderson, Leonie Jacobs

**Affiliations:** School of Animal Sciences, Virginia Tech, Blacksburg, VA 24061, USA; drewc1@vt.edu (A.M.C.); mallory8@vt.edu (M.G.A.)

**Keywords:** animal welfare, IgA, broiler chicken, environmental enrichment, stocking density

## Abstract

**Simple Summary:**

The objective of this study was to determine how environmental complexity and stocking density impacted immunoglobulin-A (IgA) concentrations in fast-growing broiler chickens. We found that birds housed in complex environments had increased plasma IgA concentrations, suggesting that these birds experienced less chronic stress than birds in simple environments. Additionally, we found that birds raised in low-density environments showed increased concentrations of secretory IgA, suggesting that these birds experienced less chronic stress than birds in high-density environments. This indicates that plasma and secretory IgA concentrations in broiler chickens do respond to housing conditions. These measures of chronic stress show appropriate contrast when the statistical power is high. However, additional work needs to replicate outcomes under similar and different conditions before these measures can be routinely used to quantify chronic stress in broiler chickens.

**Abstract:**

Commercial housing conditions may contribute to chronic negative stress in broiler chickens, reducing their animal welfare. The objective of this study was to determine how secretory (fecal) and plasma immunoglobulin-A (IgA) levels in fast-growing broilers respond to positive and negative housing conditions. In three replicated experiments, male Ross 708 broilers (*n* = 1650/experiment) were housed in a 2 × 2 factorial study of high or low environmental complexity and high or low stocking density. In experiments 1 and 3 but not in experiment 2, high complexity tended to positively impact day 48 plasma IgA concentrations. When three experiments were combined, high complexity positively impacted day 48 plasma IgA concentrations. Stocking density and the complexity × density interaction did not impact day 48 plasma IgA concentrations. Environmental complexity and the complexity × density interaction did not impact day 48 secretory IgA concentrations. A high stocking density negatively impacted day 48 secretory IgA concentrations overall but not in individual experiments. These results suggest that environmental complexity decreased chronic stress, while a high stocking density increased chronic stress. Thus, plasma IgA levels increased under high-complexity housing conditions (at day 48), and secretory IgA levels (at day 48) decreased under high-density conditions, suggesting that chronic stress differed among treatments. Therefore, these measures may be useful for quantifying chronic stress but only if the statistical power is high. Future research should replicate these findings under similar and different housing conditions to confirm the suitability of IgA as a measure of chronic stress in broiler chickens.

## 1. Introduction

The World Health Organization defines stress as any change that causes physical, emotional, or physiological strain to an animal [1]. Stress activates the hypothalamus–pituitary–adrenal (HPA) axis, which initiates a chain of signaling that culminates in the release of glucocorticoids [2]. Stress can be positive (eustress) or negative (distress) and short-term (acute) or long-term (chronic). From an animal welfare perspective, it is desirable to avoid acute and chronic distress and promote opportunities for acute and chronic eustress. 

Associations between environmental factors and stress are well established in broiler chickens, for whom barren housing conditions [3,4,5,6,7,8], high stocking densities [9,10,11,12,13,14,15,16,17] and excessive heat [18] may cause acute and chronic physical or emotional distress. For example, barren housing conditions can lead to negative affective states, reduced frequencies of natural behaviors such as perching or dustbathing, and frustration and boredom [19,20,21,22]. High stocking density can lead to decreased foot and leg health [9,11,13,16,23,24], frustration and boredom [19,20], and increased acute distress, which is reflected in increased concentrations of circulating glucocorticoids and heat shock protein 70 [25,26]. 

In production animals, stress is often quantified using circulating glucocorticoids [25,26], which are steroid hormones produced in the adrenal gland. Circulating glucocorticoids are reflective of experienced acute stress and indicate arousal (intensity) rather than the valence (positivity/negativity) associated with a stimulus [27,28]. Therefore, interpretations of the impacts of housing conditions on circulating glucocorticoids are difficult as similar increases in glucocorticoid concentrations could occur from both positive and negative stimuli as long as their intensities were comparable. For example, pigs subjected to a dominance intruder test showed nearly identical glucocorticoid responses between winners and losers [29]. In this instance, the dominance test was a negative experience for the losers and a positive experience for the winners; however, this difference is impossible to evaluate from glucocorticoid concentrations alone. Therefore, it is not possible to evaluate the valence of stimuli when only assessing the glucocorticoid response. This limits the usefulness of glucocorticoid measures to only inherently negative stimuli, such as catching and loading for transportation in broiler chickens, as the valence of the stimuli is known, and glucocorticoids can be used to measure the intensity [30,31]. Additionally, circulating glucocorticoids are only suitable as measures of acute rather than chronic stress as concentrations of blood glucocorticoids can change within minutes [32]. Another commonly used measure of stress in production animals is the H:L ratio [33]. However, the interpretation of the H:L ratio faces many of the same issues as glucocorticoids [33]. Measurements of H:L ratios have been criticized as several system-specific factors can impact H:L ratios regardless of stress levels [33,34]. Additionally, the sensitivity of the H:L ratio to the valence of housing conditions is unclear. Therefore, there is a need for indicators for chronic stress that can reflect the valence of stimuli. 

Secretory and plasma immunoglobulin-A (IgA) concentrations show potential as measures of chronic stress. IgA is the most common antibody found on mucosal surfaces and functions as a first line of defense against inhaled and ingested pathogens [35,36,37]. While both secretory IgA (SIgA) and plasma IgA (PIgA) function primarily as parts of the innate immune system, concentrations of SIgA and PIgA are independent due to differences in their structures and production. SIgA is dimeric and is produced by resident mature B-cells at mucosal surfaces, while PIgA is monomeric and produced by B-cells in bone marrow [38]. Additionally, SIgA contains the polymeric immunoglobulin receptor (PIgR) as a secretory component that could control the production of SIgA [35,37,39].

Concentrations of both SIgA and PIgA are downregulated in response to physiological or physical stress and upregulated in situations of positive experiences [37,40,41,42,43,44,45]. In rats and pigs, transferring animals to metabolic housing systems decreased SIgA concentrations compared to animals in non-metabolic housing systems [46,47,48]. In rats [39,49], mice [50], and horses [51], forced intense exercise decreased SIgA and serum IgA concentrations, while voluntary exercise had the opposite effect. When a male rat is paired with a female, the male’s SIgA concentrations initially decrease before steadily increasing, indicating that male/female pairing is a positive stimulus [43]. The opposite occurred when male rats were grouped with five other males, with SIgA concentrations steadily decreasing, indicating that male-only group housing is a negative stimulus [43]. 

Few studies have investigated the response of SIgA concentrations in chickens to environmental stimuli. Chronic heat stress decreased SIgA concentrations in Cobb and Qing Yuan Ma broilers and Hy-line Brown laying hens [18,52]. Additionally, Bovan Brown laying hens housed in enriched floor pens showed increased SIgA concentrations compared to hens raised in conventional battery caging [41]. The impact of chronic housing stressors other than heat stress on broiler chickens’ IgA concentrations is unclear. 

Environmental complexity can improve broiler chicken welfare outcomes, such as improving affective states [22], health [20,53,54], and the occurrences of species-specific behaviors [20]. Environmental complexity decreased anxiety and fear and increased optimism in Ross 708 broiler chickens compared to birds raised in more barren housing conditions [21,22]. Environmental complexity benefitted leg and bone health, decreasing lameness as broilers aged [53,54]. In Japanese quail, environmental enrichments decreased the negative impacts of a chronic stressor (repeated restraint stress) on the immune system [55]. While most studies report a positive impact of environmental complexity on broiler chicken welfare outcomes, some studies report no impact [56,57,58,59]. 

High stocking density is a well-studied housing condition for broiler chickens that can negatively impact welfare outcomes including behavior, performance, and leg and foot health [9,10,11,12,13,14,15,16,17]. High densities (40 kg/m^2^) increased gait scores (more severe lameness) compared to birds raised at lower densities (34 kg/m^2^) [23]. Additionally, the severity of contact dermatitis increased at high densities (34–36 kg/m^2^) compared to lower densities (32 kg/m^2^) [60]. High densities can cause the disturbance of rest [14,16,23], injuries [15], pain [15], distress [61], mortality [15], and carcass damage [15].

The sensitivities and responses of SIgA and PIgA concentrations to positive and negative stimuli in chickens and other species indicate their potential applications as measures of chronic stress in broiler chickens. However, the impacts of housing conditions other than heat stress, such as complexity and stocking density, have not yet been examined. Additionally, the potential combined impacts of environmental complexity and stocking density on PIgA and SIgA are unknown. Therefore, the objective of this study was to determine how environmental complexity and stocking density impacted SIgA and PIgA concentrations in fast-growing broiler chickens. We hypothesized that birds housed in high-complexity environments would have increased IgA concentrations, and birds housed in high-density environments would have decreased IgA concentrations. We hypothesized that the broilers housed in highly complex, low-density environments would show higher IgA concentrations, indicating lower levels of chronic distress, compared to broilers from low-complexity, high-density environments. 

## 2. Materials and Methods

### 2.1. Animals and Housing

This experiment was approved by the Virginia Tech Institutional Animal Care and Use Committee (IACUC protocol 19–175). In three separate replicated experiments, day-old Ross 708 male broilers (*n* = 1650 birds/experiment) were sourced from a commercial hatchery (Elizabethtown, PA, USA). The birds were vaccinated for Marek’s disease at the hatchery. The experiments consisted of a 2 × 2 factorial design comparing environmental complexity (high complexity vs. low complexity) and stocking density (high density vs. low density) as factors. The birds were raised from d1 to d50 to match the normal production period in the United States broiler chicken industry [62,63]. At d1 of age, the birds were randomly allocated to 12 pens, each containing one of four treatments (high complexity/high density, high complexity/low density, low complexity/high density, and low complexity/low density), with three replicates per treatment in a randomized block design. 

Each pen (14.5 m^2^) contained clean pine wood shavings (~10 cm depth), 4 steel galvanized feeders, and 3 nipple water lines. The birds were phase-fed a commercial corn–soy broiler diet which was formulated to meet their nutritional requirements and included a starter phase, a grower phase, and a finisher phase. The birds had ad libitum access to feed and water. During the first week, the pens contained 3 heat lamps and were under continuous lighting. Due to a technical problem in experiment 1, the birds received 24 h light for an additional week from d7 to 14. Thereafter, the lighting schedule was 18L:6D, with a light intensity of ~15 lux during light hours. The temperature was 35 °C on d1, gradually reduced to 21 °C on d24, and maintained until d50. Due to infectious bronchitis in experiment 1, the birds were given a therapeutic dose of antibiotics from d33 to d40 of age via the water lines (morbidity approximated at ~100%; the mortality due to the pathogen was 3.6%).

### 2.2. Environmental Complexity

The high-complexity pens contained four functional spaces with a “feeding” area, a “comfort” area, a “resting” area (all ~3.2 m^2^), and an “exploration” area (~4.3 m^2^). The feeding area contained all feeders and one-third of a mineral pecking stone broken into smaller pieces (Proteka, Inc., Lucknow, ON, Canada). The comfort area contained a wooden dust bath (~2 m^2^) with ~68 kg of playground sand (QUIKRETE, Atlanta, GA, USA), which was raked and partially replaced during rearing. The resting area contained three perching structures [22]. In experiment 1, perches (183 cm L × 31 cm W × 9 cm H) were built out of 1.9 cm diameter PVC pipe and were treated with textured black spray paint (Rust-Oleum, Vernon Hills, IL, USA). After anecdotal observations of limited perch use in experiment 1, we adjusted the perch design for experiments 2 and 3. Perching platforms (122 cm L × 46 cm W × 8 cm H) were constructed from 10 cm wide wooden boards. In experiment 1, the linear perching space per bird was 15.2 cm in the low-density pens and 7.6 cm in the high-density pens. In experiments 2 and 3, the perching space per bird was 76 cm^2^ in the low-density pens and 39 cm^2^ in the high-density pens. In the exploration area, six temporary enrichments were paired and rotated on a three-day schedule. These enrichments included four colored plastic balls (Clink N’ Play, Bellevue, WA, USA), four yellow treat dispenser balls with oats (Lixit Corp., Napa, CA, USA), four polyethylene string bundles suspended at bird level, iceberg lettuce (experiment 1: chopped and placed in a red rubber Kong toy (KONG, Golden, CO, USA) on litter) or cabbage (experiments 2 and 3: half a head suspended at bird level), four metal wire balls filled with alfalfa hay (Darice, Strongsville, OH, USA), and laser lights manually projected in the pens two times a day for 5 min. The enrichments were paired so that each set contained a nutritional and occupational enrichment: hay balls/hanging strings, yellow oat balls/plastic balls, and lettuce or cabbage/laser lights. 

The low-complexity pens were also divided into four functional areas but contained no enrichments. Four galvanized steel feeders were dispersed throughout the pen. Detailed pen design illustrations and photos can be found in [22]. 

### 2.3. Stocking Density

The high-density pens contained 180 birds and the low-density pens contained 90 birds per pen for final targeted stocking densities of 40–42 kg/m^2^ and 20–22 kg/m^2^, respectively. The final stocking densities at d50 are presented in Table 1. 

### 2.4. Measurements

Blood samples were collected at d28 and d48 of age in all three experiments. Blood was collected from five birds per pen (15 samples/treatment/experiment) via the brachial vein and transferred into glass collection tubes coated with 0.5% EDTA solution, gently inverted a few times, placed on ice, and then transported to the laboratory for further processing. Following transport, the blood samples were centrifuged at 10,000× *g* for 5 min. Plasma was separated from red blood cells before storage at −20 °C. During blood collection, the sample times were recorded from the initiation of handling to the removal of the needle from the bird to ensure that all samples were collected in under two minutes.

Fresh fecal samples were collected from the pen floors on d48 of age in all three experiments following observations of defecation. The samples were stored in microcentrifuge tubes, placed on ice, and then transported to the lab for storage at −80 °C until further processing. In the first two experiments, the fecal samples were pooled at the pen level following collection, resulting in 12 samples collected per experiment (3 samples/treatment/experiment). In experiment 3, individual fecal samples were collected (*n* = 5/pen) from arbitrary individuals following visual confirmations of defecation to increase the sample size.

For the quantification of SIgA, the total protein content of the fecal samples was extracted via a saline extraction method [41,47,64,65]. In short, the samples were weighed and suspended in an extraction buffer of 0.01 phosphate buffered saline, 0.5% tween (Sigma-Aldritch, St. Louis, MO, USA), and 0.05% sodium azide at a ratio of 10 mL of buffer to 1 g of sample. Following suspension, the samples were centrifuged at 1500× *g* for 20 min at 5 °C. The sample supernatant was then separated from solids, transferred into a microcentrifuge tube with 20 µL of a protease inhibitor cocktail (Sigma Aldrich., St. Louis, MO, USA), and homogenized before storage at −20 °C. 

The concentrations of SIgA and PIgA (µg/µL) were determined using a commercial enzyme-linked immunosorbent assay (ELISA; Abcam, Cambridge, MA, USA), following the manufacturer’s instructions. Intra-assay CV% values were below 2% for all samples (min 0.2%; max 2%). During sampling, it was not feasible to blind the sample collectors for treatment as the samples were collected in the poultry facility. However, during sample analysis, laboratory technicians were blinded for treatments. 

### 2.5. Statistical Analysis

All statistical analyses were performed in JMP Pro 16 (SAS Institute Inc., Cary, NC, USA). The pen was considered the experimental unit and the bird the observational unit for the PIgA analysis. In experiments 1 and 2, the pen was the experimental and observational unit for SIgA analysis, while in experiment, the 3 pen was the experimental unit and the sample (*n* = 5) was the observational unit. Data residuals for all response variables were normally distributed based on a visual inspection of the normal quantile plots. Data from each experiment were analyzed per experiment using linear mixed models with environmental complexity, stocking density, and their interactions as fixed factors. The pen number was included as a random factor. In addition, data from all three experiments were combined and analyzed using linear mixed models with environmental complexity, stocking density, and their interactions as fixed factors. The pen numbers and the experiment number were included as random factors. A post-hoc analysis was performed with Tukey HSD corrections. Associations were considered significant at *p* ≤ 0.05 and trends were identified at *p* ≤ 0.1. All data are presented as LSmeans ± SEM. 

## 3. Results

### 3.1. Day 28 Plasma IgA Concentrations

The d28 PIgA concentrations were impacted by a complexity × density interaction (F_1,59_ = 6.20; *p* = 0.037) in experiment 2 and tended to be impacted by the interaction when all three trials were combined (F_1,179_ = 1.93; *p* = 0.055; Table 2), but this was not the case in experiments 1 (F_1,59_ = 0.02; *p* = 0.894) or 3 (F_1,59_ = 0.549; *p* = 0.479). The pairwise comparisons in experiment 2 and overall were not significant (*p* > 0.1). The complexity or stocking density treatments did not impact d28 PIgA concentrations in experiment 1 (complexity: F_1,59_ = 0.87, *p* = 0.378; stocking density: F_1,59_ = 0.59, *p* = 0.465), experiment 2 (complexity: F_1,59_ = 1.33; *p* = 0.276; stocking density: F_1,59_ = 0.08; *p* = 0.789), or experiment 3 (complexity: F_1,59_ = 0.04, *p* = 0.845; stocking density: F_1,59_ = 0.234, *p* = 0.642; Table 2). Overall, the d28 PIgA did not differ due to the complexity (F_1,179_ = 0.41; *p* = 0.525) or stocking density treatments (F_1,179_ = 0.289; *p* = 0.591; Table 2).

### 3.2. Day 48 Plasma IgA Concentrations

The d48 PIgA concentrations were not impacted by the complexity × density interaction during experiment 1 (F_1,59_ = 1.00; *p* = 0.347), experiment 2 (F_1,59_ = 0.17; *p* = 0.695), experiment 3 (F_1,59_ = 0.234; *p* = 0.6414), or overall (F_1,179_ < 0.1; *p* = 0.997). The d48 PIgA concentrations tended to be increased in the high-complexity treatment during experiment 1 (F_1,59_ = 4.49; *p* = 0.068) and experiment 3 (F_1,59_ = 3.52; *p* = 0.098) but not experiment 2 (F_1,59_ = 0.74; *p* = 0.420; Figure 1). Across the three experiments, d48 PIgA concentrations were increased in the high-complexity treatment compared to the low-complexity treatment (F_1,179_ = 6.87; *p* = 0.011; Figure 2). The stocking density did not impact the d48 PIgA concentrations in individual experiments (1: F_1,59_ < 0.01, *p* = 0.991; 2: F_1,59_ = 0.34, *p* = 0.579; 3: F_1,59_ = 0.22, *p* = 0.646; Figure 1) or overall (F_1,179_ = 0.19; *p* = 0.664; Figure 2).

### 3.3. Day 48 Fecal IgA Concentrations

No impact of the complexity × density interaction on d48 SIgA was observed in experiment 1 (F_1,11_ = 0.40; *p* = 0.544), experiment 2 (F_1,11_ = 0.056; *p* = 0.819), experiment 3 (F_1,59_ = 0.96; *p* = 0.357), or overall (F_1,59_ = 0.01; *p* = 0.947). D48 SIgA concentrations were not impacted by the environmental complexity or stocking density treatments in experiment 1 (complexity: F_1,8_ = 0.17, *p* = 0.694; stocking density: F_1,8_ = 1.59, *p* = 0.247), experiment 2 (complexity: F_1,8_ = 2.29, *p* = 0.168; stocking density: F_1,8_ = 2.08, *p* = 0.187), or experiment 3 (complexity: F_1,59_ = 0.03, *p* = 0.872: stocking density: F_1,59_ = 0.90, *p* = 0.371). Environmental complexity did not impact d48 SIgA concentrations across all three experiments (F_1,59_ = 0.15; *p* = 0.699; Figure 3). Overall, d48 SIgA concentrations were decreased in the high-density treatment compared to the low-density treatment (F_1,83_ = 4.69; *p* = 0.033; Figure 3). 

## 4. Discussion

This study investigated the impacts of positive (environmental complexity) and negative (stocking density) housing conditions on broiler chicken PIgA and SIgA concentrations in three replicated experiments. We did not confirm the hypothesis that broilers housed in highly complex, low-density environments would show higher IgA concentrations indicative of lower levels of chronic stress compared to broilers in low-complexity, high-density environments. We were able to confirm our hypothesis that IgA concentrations increased in response to high-complexity environments and decreased in response to high-density environments as the PIgA and SIgA concentrations on day 48 differed in response to environmental complexity and stocking density, respectively. This is in line with previous studies in which IgA concentrations responded to positive and negative experiences in mice, rats, dogs, horses, and laying hens [39,41,43,45,46,47,48,49,50,51,66,67]. The current study is the first to detect an impact of environmental complexity on PIgA concentrations and an impact of stocking density on SIgA concentrations in broiler chickens. 

At 28 days of age, an interaction effect between complexity and stocking density on PIgA concentrations was found in experiment 2 and across all experiments; however, no pairwise differences were found, making the interpretation of this interaction difficult. The sample sizes for this experiment were relatively low (*n* = 15 samples/treatment/experiment; *n* = 45 samples/treatment total). It is possible that this interaction would have resulted in interpretable results with a greater sample size and thus with a higher statistical power. 

Neither environmental complexity nor stocking density impacted the d28 PIgA concentrations in any of the three experiments or overall, which was unexpected. A high stocking density is considered a negative stimulus for broiler chickens [9,10,11,12,13,14,15,16,17]. The negative impacts of high stocking density generally do not occur until densities reach over 30 kg/m^2^ [10,24]. This higher density is reached when the birds reach a certain weight (as the birds grow and the space availability remains the same), which in our study, was at 33 days of age in the high-density pens. This means that the high-density birds in our study were exposed to high densities and the associated potential distress for approximately two weeks of life (day 33-day 48). At 28 days of age, the actual stocking density was still relatively low (~22–24 kg/m^2^ in the high-density pens), so it may not be experienced as a negative stressor in young broilers under our experimental conditions. In line, the birds were less fearful when raised in high-density pens when compared to the birds raised in low-density pens at 26 days of age [22]. While stocking density is generally considered a negative stimulus [9,10,11,12,13,14,15,16,17], other studies do not report a negative impact of stocking density on broiler affective state and acute distress [21,22,68]. Future studies should investigate the impacts of stocking density on birds at different ages to determine if the length of exposure to high density impacts welfare outcomes and IgA concentrations. 

We observed a positive impact of a complex environment on the d48 PIgA concentrations in experiments 1 (trend), 3 (trend), and overall, although no association was found in experiment 2. The tendencies observed in experiments 1 and 3 may be due to a relatively small sample size, which is supported by the statistical association found when combining the three experiments, and the numerical (non-significant) differences observed in experiment 2. Repeatability is important for potential measures of chronic stress as measures must first be consistent between flocks under similar housing conditions before they can be used to compare flocks in different housing systems. The inclusion of multiple replicated experiments within this study allowed for the comparison of the impact of similar housing conditions across three flocks. While statistical differences between treatments were not observed in every experiment, the overall positive numerical effects of a complex environment on d48 PIgA concentrations were replicable across experiments. Future experiments investigating PIgA concentrations in relation to housing conditions should ensure a higher statistical power than the current study to confirm this finding. Additionally, our results are consistent with findings in other species [39,50,66,69,70]. A complex environment allows broilers to express species-specific behaviors, which can induce a positive affective state compared to broilers raised in low-complexity environments [20,21,22,56,71]. A low-complexity environment can result in chronic and acute distress due to the inability to display these natural behaviors. This distress could increase circulating glucocorticoid concentrations and HPA axis activity. Physiological stress responses can impact IgA concentrations via glucocorticoid signaling [37,72,73]. Glucocorticoids bind to IgA-secreting B-cells and reduce the expression of IgA-encoding mRNA [37,74,75,76]. A decreased expression of IgA-encoding mRNA could decrease the transcription of IgA proteins. The reduced transcription of IgA proteins leads to decreased concentrations of functional IgA. Low-complexity and barren environments can cause distress [20,56,77,78,79]; thus, birds raised in these environments could have higher concentrations of circulating glucocorticoids in comparison to high-complexity environments. Increased glucocorticoid concentrations in birds housed in low-complexity environmetns results in increased interactions between glucocorticoids and IgA-secreting B-cells, thereby decreasing the expression of IgA-encoding mRNA and the transcription of IgA proteins in low-complexity birds, resulting in lower PIgA concentrations. Our results indicate that the d48 PIgA is responsive to positive housing conditions (environmental complexity) and could be useful as a measure of chronic stress in broilers, yet it is still not fully understood what role glucocorticoids play in the production of PIgA, especially when broilers are kept in positively valenced conditions.

Environmental complexity did not impact d48 SIgA concentrations. The lack of impact may indicate that environmental complexity was not a positive housing stimulus for broiler chickens. However, this is contradictory to our d48 PIgA results and previous studies, which indicated that environmental complexity reduced distress and improved affect in broilers [20,21,22,54,56,80]. For example, broilers housed in high-complexity environments were less fearful, less anxious, and less fearful following acute stressors than broilers housed in low-complexity environments [22,80]. Additionally, this is contradictory to previous studies which show that SIgA responds to positive experiences in other species [39,50]. In broiler chickens, SIgA may be insensitive to positive experience, which cannot be confirmed by previous research. It is also possible that we did not detect an impact of environmental complexity due to a measurement error. Fecal IgA can be difficult to quantify as fecal proteases can break down SIgA quickly following defecation [41,47,64,65]. We ensured sample freshness via visually confirming defecation and freezing the samples quickly after collection. Currently, we have no evidence supporting that d48 SIgA is a useful indicator of positive experience in broilers. 

A high stocking density resulted in low d48 SIgA concentrations when all experiments were combined. A high stocking density can negatively impact broiler health [11,81], behavior [13,23], and can cause distress [68,82,83]. In this context, decreased SIgA concentrations indicate increased chronic stress in broilers housed in high-density pens compared to the broilers in low-density pens. One proposed mechanism for this is that high levels of circulating glucocorticoids in distressed animals interact with PIgR-secreting epithelial cells at mucosal surfaces and decrease PIgR concentrations in the mucus [44,50,72,73,76,84,85]. PIgR is the secretory component of SIgA and is required for SIgA to cross epithelial barriers and enter the mucosal lumen [37,86]. If PIgR concentrations are decreased in distressed animals, SIgA concentrations are also decreased as less SIgA can be released into mucosal lumens. The d48 SIgA concentrations responded to high stocking density as hypothesized; therefore, they can be a viable measure of chronic negative stress (high stocking density) in broiler chickens. We recommend replication under similar and different conditions to confirm the viability of this chronic distress measure in broilers.

The negative impact of high density was not reflected in the d48 PIgA concentrations, in which no impact of high density was observed. This result may be due to the different production mechanisms for PIgA and SIgA [86]. Circulating PIgA is monomeric and does not bind with PIgR to enter circulation, so glucocorticoids can only reduce PIgA production via direct interactions with IgA-secreting B-cells in bone marrow [86], limiting the impact of these hormones on PIgA. However, SIgA is dimeric and requires the inclusion of PIgR to enter the mucosal lumen [37,86]. PIgR-secreting epithelial cells can also interact with glucocorticoids under chronic stress, decreasing PIgR production and indirectly decreasing SIgA concentrations. Therefore, glucocorticoids can interact with two mechanisms to decrease SIgA concentrations (direct interactions with IgA secreting B-cells and PIgR secreting epithelial cells) while only being able to impact PIgA via direct interactions with IgA-secreting B-cells. These mechanisms suggest that SIgA concentrations may decrease more quickly in response to increased HPA axis activity than PIgA concentrations, which are less accessible to circulating glucocorticoids. This explains why SIgA concentrations were lowered in response to the high-density treatment but PigA concentrations were not, as the birds may have only experienced the high-density treatment as negative for the last few weeks of life. Overall, these results indicate that d48 SIgA is sensitive to chronic stress caused by high stocking density while PIgA concentrations are not. 

A limitation to the interpretation of the results from experiment 1 is that the birds experienced a period of continuous lightning during their second week of life and had an infectious bronchitis exposure which required treatment with antibiotics from d33-d40. Continuous light exposure [87] and pathogen exposure [88] could have impacted plasma and secretory IgA concentrations. Treatment with antibiotics can decrease IgA concentrations and lead to IgA deficiency with long-term broad spectrum antibiotic use [89] but not with the short-term use of antibiotics similar to the treatment applied in this study [90,91]. In line, the d48 PIgA concentrations in experiment 1 and 2 were very similar (Figure 1), suggesting that these two factors might not have impacted PIgA concentrations at time of sampling.

To our knowledge, no studies have compared how PIgA and SIgA respond to circulating glucocorticoid concentrations in poultry. This is the first study to assess both PIgA and SIgA in response to housing conditions in non-human animals, which makes it difficult to confirm our proposed theories. Therefore, we recommend further investigation into the differences in production mechanisms between PIgA and SIgA to determine how they respond to circulating glucocorticoid concentrations under similar and different housing conditions. 

The d48 PIgA concentrations varied largely between experiments, which suggests that PIgA concentrations may not be consistent between flocks. Concentrations of physiological measures of chronic stress should ideally be consistent between flocks so that the welfare measure allows for direct comparisons between flocks and studies. The variation in d48 PIgA concentrations suggests it may not be possible to determine “normal” PIgA concentrations. Instead, PIgA may be more appropriate for use as a relative comparative indicator between treatments in a single experiment or between similar husbandry conditions in non-experimental contexts. 

This is the first study to investigate the impacts of housing conditions on PIgA and SIgA concentrations in broiler chickens. While the sample sizes were relatively low, especially for SIgA during experiments 1 and 2 (*n* = 12/experiment), we were able to detect differences between complexity treatments for PIgA and between stocking density treatments for SIgA. This indicates that plasma IgA concentrations on day 48 show promise as a biomarker for positive experiences in broiler chickens, and secretory (fecal) IgA concentrations show promise as a biomarker for negative experiences in broiler chickens.

## 5. Conclusions

Environmental complexity positively impacted plasma immunoglobulin-A concentrations in broiler chickens at 48 days of age, suggesting that levels of chronic stress were reduced when the chickens were housed in highly complex environments compared to simple environments. This difference was not observed in plasma immunoglobulin-A concentrations at 28 days of age or secretory (fecal) immunoglobulin-A concentrations at day 48. High stocking density negatively impacted secretory (fecal) immunoglobulin-A concentrations in broiler chickens at 48 days of age across all three experiments, suggesting the chickens experienced increased levels of chronic stress when housed under high-density conditions compared to low-density conditions. This difference was not observed for plasma immunoglobulin-A concentrations at day 28 or day 48. Our results show the potential of the day 48 plasma immunoglobulin-A concentration as an indicator of chronic eustress as it increased in response to complexity, which is considered a positively valenced stimulus. Furthermore, our results show the potential of the day 48 secretory (fecal) immunoglobulin-A concentration as an indicator for chronic distress as it decreased in response to a high stocking density, which is considered a negatively valenced stimulus. Future research should attempt to replicate these outcomes under similar and different conditions to confirm these factors as measures of animal welfare for broiler chickens. 

## Figures and Tables

**Figure 1 animals-13-02058-f001:**
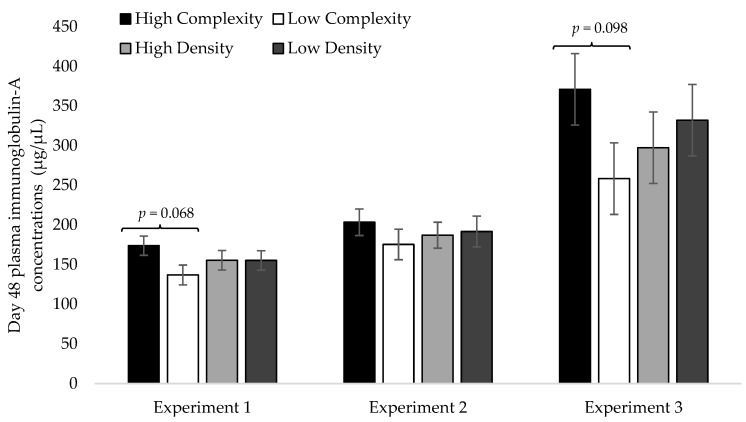
Least square mean estimates (±SEMs) of plasma immunoglobulin-A concentrations in broilers at 48 days of age housed at high or low complexity and at high or low stocking density in experiment 1 (*n* = 60), experiment 2 (*n* = 60), and experiment 3 (*n* = 60).

**Figure 2 animals-13-02058-f002:**
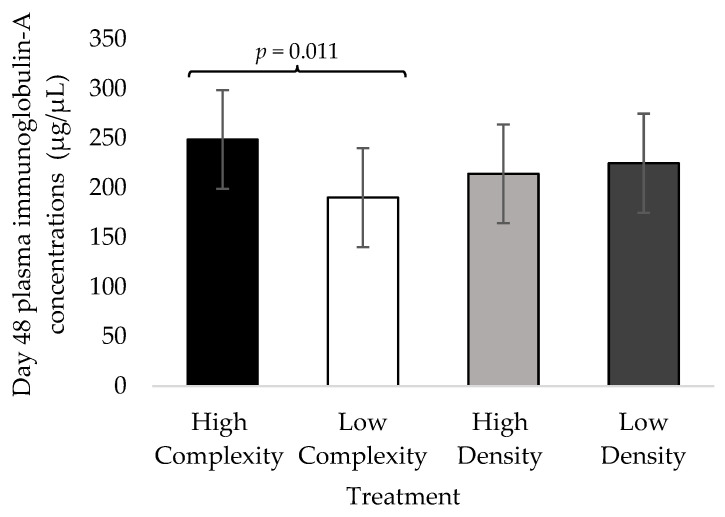
Least squared mean estimates (±SEMs) of plasma immunoglobulin-A concentrations in broilers at 48 days of age housed at high or low complexity and at high or low stocking density across all three experiments (*n* = 180).

**Figure 3 animals-13-02058-f003:**
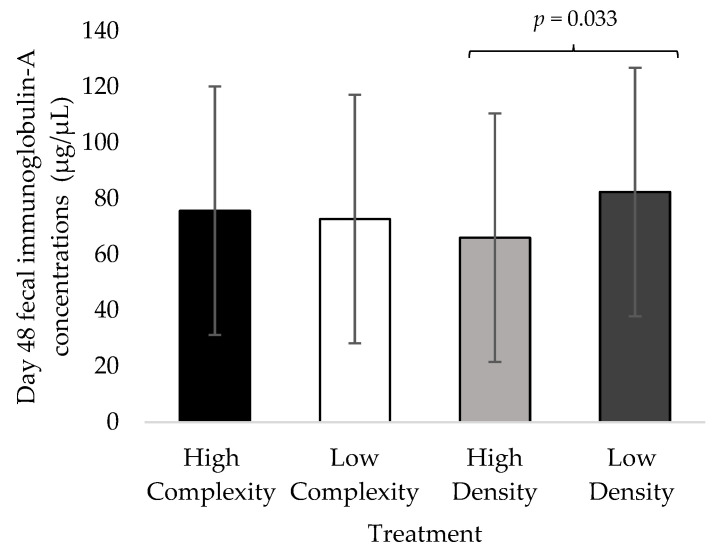
Least squared mean estimates (±SEMs) of secretory (fecal) immunoglobulin-A concentrations in broilers at 48 days of age housed at high or low complexity and at high or low stocking density across all three experiments (*n* = 90).

**Table 1 animals-13-02058-t001:** Raw means of day 50 pen stocking densities in experiment 1, experiment 2, and experiment 3 for both stocking density treatments.

Experiment	Day 50 Stocking Density (kg/m^2^)	
	High-Density Pens	Low-Density Pens
1	42.1	23.8
2	42.6	23.3
3	42.1	22.1

**Table 2 animals-13-02058-t002:** LSmean ± SEM estimates of day 28 plasma immunoglobulin-A (µg/µL plasma) concentrations in experiment 1 (*n* = 60), experiment 2 (*n* = 60), experiment 3 (*n* = 60), and overall (*n* = 180).

	Plasma Immunoglobulin-A (µg/µL) on Day 28 of Age			
Treatment	Experiment 1	Experiment 2	Experiment 3	Overall
High complexity (HC)	139 ± 12	265 ± 31	213 ± 19	208 ± 29
Low complexity (LC)	156 ± 12	215 ± 31	219 ± 19	196 ± 29
High density (HD)	155 ± 12	246 ± 31	223 ± 19	207 ± 29
Low density (LD)	141 ± 12	234 ± 31	209 ± 19	197 ± 29
HC/HD	145 ± 18	327 ± 44	210 ± 28	231 ± 22
HC/LD	134 ± 18	204 ± 44	217 ± 28	184 ± 21
LC/HD	164 ± 18	165 ± 44	236 ± 28	176 ± 21
LC/LD	148 ± 18	264 ± 44	202 ± 28	208 ± 22

## Data Availability

Data are available from the corresponding author upon reasonable request.

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
