# Peer review of "Measuring Chronic Stress in Broiler Chickens: Effects of Environmental Complexity and Stocking Density on Immunoglobulin-A Levels"

_animals, 2023, doi:10.3390/ani13132058_

Round 1
Reviewer 1 Report
This study investigates the effect of environmental complexity (Enriched/barren) and housing density (low/high) on fecal and plasma immunoglobulin-A (IgA) levels in fast-growing broilers. It is interesting to see the possibility of using these traits as measures of chronic stress which identify the valence associated with environmental conditions. The paper is well written and has novel information that adds to our knowledge.
Measuring Chronic Stress in Broiler Chickens: Effects of Environmental Complexity and Stocking Density on Immunoglobulin-A Levels
This study investigates the effect of environmental complexity (Enriched/barren) and housing density (low/high) on fecal and plasma immunoglobulin-A (IgA) levels in fast-growing broilers. It is interesting to see the possibility of using these traits as measures of chronic stress which identify the valence associated with environmental conditions. The paper is well written and has novel information that adds to our knowledge.
1. What is the main question addressed by the research?
The main question was if plasma and fecal IgA would be suitable indicators for chronic stress and if they would be associated with the response of birds to positive or negative environmental conditions.
2. Do you consider the topic original or relevant in the field? Does it address a specific gap in the field?
Yes. The topic is original. It expands our knowledge by addressing the potential for using IgA levels as a measure of chronic stress in broilers.
3. What does it add to the subject area compared with other published material?
Glucocorticoids are largely used as a physiological indicator of stress but it may be difficult to interpret the results under chronic stress conditions. Besides, it is important to find out a measure which can be associated with the responses of birds differing with positive or negative environmental conditions that may also be associated with the emotional state of birds. The results from overall data suggested that increased stocking density (negative environment) reduced fecal IgA while enrichment (positive environment) increased plasma IgA levels. So, I found the study approach interesting and the results worth studying further.
4. What specific improvements should the authors consider regarding the methodology? What further controls should be considered?
The study relies on three replicated experiments and it improves the strength. However, in future research, measuring some other additional traits such as H/L ratio would be helpful to interpretation of the results.
5. Yes.
6. Yes.
7. They are clear. No additional comment.
Author Response
Thank you for your time and effort to review this manuscript and provide valuable feedback. We have provided detailed responses to each comment below.
Measuring Chronic Stress in Broiler Chickens: Effects of Environmental Complexity and Stocking Density on Immunoglobulin-A Levels
This study investigates the effect of environmental complexity (Enriched/barren) and housing density (low/high) on fecal and plasma immunoglobulin-A (IgA) levels in fast-growing broilers. It is interesting to see the possibility of using these traits as measures of chronic stress which identify the valence associated with environmental conditions. The paper is well written and has novel information that adds to our knowledge.
- What is the main question addressed by the research?
The main question was if plasma and fecal IgA would be suitable indicators for chronic stress and if they would be associated with the response of birds to positive or negative environmental conditions.
- Do you consider the topic original or relevant in the field? Does it address a specific gap in the field?
Yes. The topic is original. It expands our knowledge by addressing the potential for using IgA levels as a measure of chronic stress in broilers.
- What does it add to the subject area compared with other published material?
Glucocorticoids are largely used as a physiological indicator of stress but it may be difficult to interpret the results under chronic stress conditions. Besides, it is important to find out a measure which can be associated with the responses of birds differing with positive or negative environmental conditions that may also be associated with the emotional state of birds. The results from overall data suggested that increased stocking density (negative environment) reduced fecal IgA while enrichment (positive environment) increased plasma IgA levels. So, I found the study approach interesting and the results worth studying further.
- What specific improvements should the authors consider regarding the methodology? What further controls should be considered?
The study relies on three replicated experiments and it improves the strength. However, in future research, measuring some other additional traits such as H/L ratio would be helpful to interpretation of the results.
The reviewer brings up a good point. We have considered including glucocorticoid or H:L ratios as ‘standard’ stress measures to compare to IgA concentrations. However, both glucocorticoid and H:L ratios have been criticized for their lack of repeatability and reliability. We discuss some of those problems with glucocorticoids in the manuscript. H:L ratios face many of the same issues as glucocorticoids. In short, glucocorticoid concentrations are indicative only of the intensity of a stimulus, not the valence (Moberg, 2000 https://www.cabi.org/vetmedresource/ebook?ebook=20002215200; Mendl et al. 2010 https://royalsocietypublishing.org/doi/10.1098/rspb.2010.0303). Therefore, utilizing glucocorticoid concentrations as a standard for a study investigating the impacts of housing systems with different valences would not be appropriate (i.e. the complexity could be experienced just as intense as high density, resulting in ‘the same’ CORT responses). Additionally, glucocorticoid concentrations are indicative of acute stress (Romero et al. 2005, https://pubmed.ncbi.nlm.nih.gov/15664315/), while IgA concentrations are assessed as an indicator of chronic stress. Therefore, comparisons of corticosterone and IgA concentrations wouldn’t add value, as animals can experience chronic and acute stresses of different valences simultaneously. However, while H:L ratio has been used as a measure of chronic stress in birds, the sensitivity of H:L ratio to the valence of housing systems is unclear. Additionally, H:L ratios can be impacted by several system-specific factors regardless of stress level (Lentfer et al. 2015; https://pubmed.ncbi.nlm.nih.gov/25622692/). These results are similar to what is found by Skwarska in her systematic review of H:L ratios in Great Tits (2019; https://bioone.org/journals/acta-ornithologica/volume-53/issue-2/00016454AO2018.53.2.001/Variation-of-Heterophil-to-Lymphocyte-Ratio-in-the-Great-Tit/10.3161/00016454AO2018.53.2.001.full).
As limitations of glucocorticoids were already discussed, we added some of these limitations of H:L ratios to the introduction to: “Another commonly used measure of stress in production animals is H:L ratio [33]. How-ever, interpretation of H:L ratios face many of the same issues as glucocorticoids [33]. Measurement of H:L ratios have been criticized, as several system-specific factors can impact H:L ratios regardless of stress level [33,34]. Additionally, the sensitivity of H:L ratios to the valence of housing conditions is unclear.” L71-75
- Yes.
- Yes.
- They are clear. No additional comment.
Reviewer 2 Report
Excellent presentation. No major comments from my side.
Only, the materials and methods are a bit complicated. Maybe a visual description, some photos, and less numbers could help.
Author Response
Thank you so much for your time and effort in reviewing our manuscript and your positive responses. We appreciate your time and expertise.
Excellent presentation. No major comments from my side.
Only, the materials and methods are a bit complicated. Maybe a visual description, some photos, and less numbers could help.
The experimental design was previously described in Anderson et al. (2021a and 2021b; https://doi.org/10.1038/s41598-021-95280-4 and https://doi.org/10.3390/ani11082383) including pictures and illustrations of pen designs. We made some edits throughout the methods section for clarity and moved the stocking density information to a table to improve readability. We have clarified that pictures/illustrations of pen design can be found in the references listed: “Detailed pen design illustrations and photos can be found in [22].” L182-183
Reviewer 3 Report
It seems that the number of birds used in some experiments were too low, which impared statistical analysis and demanded more powerful methods and repetition of the entire study (as stated already in the draft). That is why I gave "average" regarding scientific soundness and overall merit.
Author Response
Thank you so much for your time and effort in reviewing our manuscript. A detailed response is given below.
It seems that the number of birds used in some experiments were too low, which impared statistical analysis and demanded more powerful methods and repetition of the entire study (as stated already in the draft). That is why I gave "average" regarding scientific soundness and overall merit.
We agree with the reviewer that the sample size is low for fecal IgA measurement in experiments 1 and 2. Based on our experience in those experiments, we increased the sample size in experiment 3 to increase statistical power. As this is a novel measure in a broiler chicken welfare context, we did not have sufficient literature to determine an appropriate sample size prior to the start of the study, which was intended to include 3 replicate experiments a priori, because each experiment could only include 12 pens in total, resulting in low treatment replicates. [no edit]
Reviewer 4 Report
This manuscript puts forward innovative ideas and explores IgA as an indicator of chronic stress. Although the intention is good, but the experimental design and implementation are not comprehensive enough, and the results lack convincing evidence, it is not recommended for publication. It is recommended to expand the sample size and to design one or more stress indicators (eg. Corticosterone, H/L etc) in the future study to verify IgA as a stress indicator, and to convince readers of the feasibility of IgA as an stress indicator. The following are some more reasons in support of the review decision.
1. Ross 708 is a commercial broiler line, the growth period up to the marketing weight is below 35 days of age. The experiment lasts from d0 to d50, even the results support the use of IGA as a chronic stress indicator, there is no practical meaning anyhow.
2. Data in experiment one should not be used if birds were sick due to the pathogen exposure. The manuscript did not mention what kind of pathogen the birds were exposed to? How sick the birds were? And what is the lost of birds?
3. Fig 2: the first two bars; figure 3: the last two bars. The error bars were rather long in these two pair comparisons, but the statistical analysis showed significant differences. It’s hard to believe the statistical significance.
4. As mentioned in the first paragraph, there are some stress indicators that have been recognized and used widely. It’s nice to explore new and better indicators, but it needs a lot of validation and verification.
Reviewer 5 Report
This study was conducted to investigate the possible use of plasma and secretory IgA as biomarkers of chronic stress in broiler chickens kept under different levels of environmental complexity and stocking density.
The introduction reviews the relevant literature and clearly states the research questions. The materials and methods are explained with sufficient details to allow for the replication of the study.
Lines 144 – 149: The technical problem and the pathogen exposure were not taken into account in the discussion. Could the lack of rest (24h of light) during the second week of experiment 1 have influenced the findings on d28 of this experiment? Could the antibiotic treatment between d33 and d40 have influenced SIgA concentrations measured on d48 in this same experiment?
Lines 157 – 163: Data from all three experiments were combined and analyzed for the overall effects of treatments on measured traits. Was the change in perches (pipe to platform) and in perching space between experiments intentional? Could this have influenced the valence of the stress to which birds were exposed?
Results: The results are clearly presented.
Are the F values reported on line 230 (the interaction in exp. 1), 233 (complexity in exp. 1) and 234 (stocking density in exp. 1) negative? To my knowledge, F-values cannot be negative. Please verify this and correct if necessary.
It is not clear to me why the authors have chosen to present the results of the three replicate experiments separately. While the findings varied between experiments, only the overall effects of treatments are of interest as no conclusions could be drawn from findings of independent replicates.
Discussion: It is clear and convincing.
The limitations of the study including the relatively low sample size and the important inter-experiment variability in measured traits (e.g., PIgA) are highlighted and discussed.
On line 326 – 330, it is stated that increased glucocorticoids reduce plasma IgA concentrations and on line 374 – 375, it is stated that glucocorticoids may not reduce plasma IgA levels. Given that plasma IgA are produced by the same B-cells located in bone marrow, why would low complexity but not high stocking density reduce their levels?
Conclusions: The conclusions are supported by the findings of the study.
Round 2
Reviewer 4 Report
The revised version is acceptable.